# Effect of a *Debaryomyces hansenii* and *Lactobacillus buchneri* Starter Culture on *Aspergillus westerdijkiae* Ochratoxin A Production and Growth during the Manufacture of Short Seasoned Dry-Cured Ham

**DOI:** 10.3390/microorganisms8101623

**Published:** 2020-10-21

**Authors:** Lucilla Iacumin, Martina Arnoldi, Giuseppe Comi

**Affiliations:** Department of Agricultural, Food, Environmental and Animal Science, University of Udine, Via Sondrio 2/a, 33100 Udine, Italy; lucilla.iacumin@uniud.it (L.I.); arnoldi.martina@spes.uniud.it (M.A.)

**Keywords:** *D. hansenii*, *L. buchneri*, *A. westerdijkiae*, biocontrol, ham

## Abstract

Recently, specific dry-cured hams have started to be produced in San Daniele and Parma areas. The ingredients are similar to protected denomination of origin (PDO) produced in San Daniele or Parma areas, and include pork leg, coming from pigs bred in the Italian peninsula, salt and spices. However, these specific new products cannot be marked as a PDO, either San Daniele or Parma dry cured ham, because they are seasoned for 6 months, and the mark PDO is given only to products seasoned over 13 months. Consequently, these products are called short-seasoned dry-cured ham (SSDCH) and are not branded PDO. During their seasoning period, particularly from the first drying until the end of the seasoning period, many molds, including *Eurotium* spp. and *Penicillium* spp., can grow on the surface and work together with other molds and tissue enzymes to produce a unique aroma. Both of these strains typically predominate over other molds. However, molds producing ochratoxins, such as *Aspergillus ochraceus* and *Penicillium nordicum*, can simultaneously grow and produce ochratoxin A (OTA). Consequently, these dry-cured hams may represent a potential health risk for consumers. Recently, *Aspergillus westerdijkiae* has been isolated from SSDCHs, which could represent a potential problem for consumers. Therefore, the aim of this study was to inhibit *A. westerdijkiae* using *Debaryomyces hansenii* or *Lactobacillus buchneri* or a mix of both microorganisms. Six *D. hansenii* and six *L. buchneri* strains were tested in vitro for their ability to inhibit *A. westerdijkiae*. The strains *D. hansenii* (DIAL)1 and *L. buchneri* (Lb)4 demonstrated the highest inhibitory activity and were selected for in situ tests. The strains were inoculated or co-inoculated on fresh pork legs for SSDCH production with OTA-producing *A. westerdijkiae* prior to the first drying and seasoning. At the end of seasoning (six months), OTA was not detected in the SSDCH treated with both microorganisms and their combination. Because both strains did not adversely affect the SSDCH odor or flavor, the combination of these strains are proposed for use as starters to inhibit OTA-producing *A. westerdijkiae*.

## 1. Introduction

Recently, a particular type of dry-cured ham has begun to be produced in different Italian areas. The product is obtained from pork legs (minimum 12 kg) that are brined and seasoned for only six months. Because it is seasoned for less than 13 months, this product cannot be branded as a protected denomination of origin (PDO) product, such as San Daniele or Parma dry-cured hams. Indeed, they are called short-seasoned dry-cured ham (SSDCH), and their production process is presented in Table 1. Maturation step is the production phase in which scheduled and controlled environmental conditions (temperature, humidity, and ventilation) and salt are used to dry fresh meat, allowing also all the enzymatic reactions to obtain the final product “prosciutto”, which can be perfectly preserved without the use of additional preservatives. The uncovered surface of the thighs inner side is covered with salt ad libitum, and the salting phase lasts one day for each kilogram of thigh meat. Then the thighs are manually or mechanically pressed to facilitate the salt penetration and to drain even the last residues of blood from the meat. Subsequently, first and second drying, pre-seasoning, and seasoning are performed. The stability of the product depends on salt and dehydration/maturation process, in fact, lowering the water activity (Aw) to ≤ 0.90 [1,2] the microbial activity is inhibited. However, molds can develop spontaneously from the mycobiota of the production environment or by contamination with spices on the hams, particularly at the pre-seasoning and seasoning steps, when excessive humidity is present due to relative humidity (R.H.) control. In addition, molds grow when there are high levels of condensation on the surface of the hams, producing mycelia and spores that rapidly spread in the air and on the apparatuses of the pre-seasoning and seasoning areas. The indigenous mold mycobiota has beneficial seasoning effects, positively contribute to the unique characteristics of the final product [3]. Indeed, these molds prevent excessive drying of the surface, improve the texture, and limit excessive hardness of the flesh due to their proteolytic activities, and they also exert antioxidant effects, contribute to color production and improve the aroma and flavor [3,4,5,6,7]. The molds on the surface of dry-cured ham have been studied in great detail [3,4,7,8,9,10,11,12,13,14,15]. *Penicillium* spp., *Aspergillus* spp., and *Eurotium* spp. are the most common molds present during all stages of ham production [4]. The succession or the prevalence of specific genera strictly depends on the temperature at which seasoning is performed, the moisture of the meat, and the relative humidity [1,14,16]. However, inadequate processing procedures can increase the development of molds that can lead to negative effects, such as the production of mycotoxins (ochratoxin A) and off-flavors and off-odors. Nevertheless, in some cases, the use of short seasoning times can cause high moisture levels on the surface of the hams, which has detrimental effects, such as the growth of ochratoxin A (OTA)-producing molds. Several authors demonstrated the presence of OTA in meat products, due to the ability to grow on the surface of meat products of numerous fungi (*Penicillium verrucosum, P. nordicum, Aspergillus ochraceus,* and *A. westerdijkiae*) during seasoning and storage producing OTA [1,13,17,18]. Recently, small-scale or artisanal lots of SSDCH or meat products were contaminated with *A. westerdijkiae* [16,19,20]. This mold can grow during the drying and seasoning steps and produces over 1 µg OTA kg^−1^ meat. *A. westerdijkiae* is included in the *Circumdati* section and is considered a new species, differing from *A. ochraceus* because of the cream color of sclerotia, failed in growth at 37 °C and greater OTA production [21,22,23]. With OTA being classified by the International Agency for Research of Cancer (IARC) as a “Group B” of carcinogenic molecules in humans [24], legal limits have been introduced for OTA concentration in foods (i.e., 1 µg kg^−1^ in meat and meat products, Italian Ministry of Health Circular no. 10-93 09/06/1999). Consequently, it is essential to advance in efficient strategies to eliminate the risk [18]. For food detoxification purposes, chemical, physical, and biological approaches have been used [25,26], but they resulted in a failure to remove OTA due to its high stability [27]. Thus, an effective strategy remains to be developed that can limit or inhibit the growth of OTA-producing molds and the consequent OTA production. The most realistic and interesting method is the use of bioprotective cultures, which is based on the antagonistic activity of autochthonous yeast and LAB species against molds [26]. Several studies [1,16,18,28,29,30,31,32] have evaluated the ability of autochthonous yeast to protect meat products against mold growth. In particular, *D. hansenii* and *Candida famata* were identified as the primary yeasts present in meat products from the early stage of seasoning to the end, when it becomes the predominant strain [3,33,34,35,36]. Consequently, *D. hansenii* is largely used as a starter culture in European sausages [33,37]. In addition, *Saccharomycopsis fibuligera* gives good results in limiting the growth and OTA production of *P. nordicum* and *A. ochraceus* in speck [1], despite it rarely being present in meat products [37]. Recently, we isolated various *Lactobacillus buchneri* strains in SSDCH surfaces without molds and hypothesized they could be used as bioprotective agents against OTA producing molds. Yeasts and LAB, due to their natural presence in meat products, were used as a first weapon for biocontrol against pathogenic microorganisms [29,30,31]. Concerning yeasts, their effectiveness in molds inhibition on meat products needs to be prudently evaluated [31,36], but the success of their usage on dry-cured ham [31] has been investigated and demonstrated both in vitro and in situ against OTA-producing molds [16,29,30,38,39]. However, these results should be improved upon, because it was recently demonstrated that some mold species such as *A. westerdijkiae* can produce more OTA than others such as *A. ochraceus* and *P. nordicum*. Considering that little is known about the bioprotective effects of LAB against molds and that the many opportunities that yeasts and LAB could have in mold inhibition, the goal of this study was to evaluate the abilities of *Debaryomyces hansenii* and *Lactobacillus buchneri* to control *A. westerdijkiae* OTA production during SSDCH production. 

## 2. Materials and Methods

The microorganisms used in this study included 6 strains of *Debaryomyces hansenii* (DIAL1, DIAL2, DIAL3, DIAL4, DIAL5, and DIAL6), selected from 90 isolates from meat products, and 6 strains of *L. buchneri*, including two freeze-dried commercial *L. buchneri* strains (LABb2 and LABb3) and 4 selected from 40 isolates from SSDCH products (Lb1, Lb2, Lb3, and Lb4). Either the *D. hansenii* or *L. buchneri* strains were previous tested in vitro for their activity against *A. ochraceus*. All of the microorganisms were stored in the Collection of the Agricultural, Food, Environmental and Animal Science Department of the University of Udine (Udine, Italy). The twelve strains were selected for potential competition against *A. westerdijkiae*. The yeast strains were maintained in malt extract modified agar (MEMA, Oxoid, Italy) medium supplemented with dextrose (1%) and peptone (1%), and the LAB were cultured in MRS agar (De Man, Rogosa, Sharpe, Oxoid, Italy). 

### 2.1. Preparation of D. hansenii and L. buchneri Inocula

The *D. hansenii* strains were grown on MEMA at 25 °C for 48–72 h. Suspensions were prepared by adding a loopful of cells to peptonized water (0.7% NaCl and 0.1% peptone in 1000 mL of water). The density of the yeast cultures was determined spectrophotometrically at an optical density at 600 nm (OD_600_) of 0.1, after which serial dilutions were prepared to obtain the concentrations used for the experiments. The *L. buchneri* strains were maintained at −80 °C in vials with MRS broth + 30% (v/v) glycerol (Oxoid, Italia). The strains were grown in MRS broth (Oxoid, Italia) at 37 °C for 24 h. Then, the suspensions were standardized to an optical density at 600 nm (OD_600_) of 0.1, and their microbial counts were evaluated. Specifically, serial dilutions were performed in peptonized water, and 0.1 mL of each dilution was spread onto MRS plates (Oxoid, Italy). The plates were incubated at 37 °C for 48 h, and the resulting colonies were counted. Each suspension contained approximately 7 log CFU (colony-forming unity) mL^−1^. A mixed suspension of both microorganisms was generated by mixing 5 mL of 7 log *D. hansenii* mL^−1^ to 5 mL of 7 log *L. buchneri* mL^−1^ to obtain a final suspension at 7 log CFU mL^−1^.

### 2.2. Preparation of the Aspergillus westerdijkiae Inoculum

The *A. westerdijkiae* strain used in this study was isolated from meat product and stored in the Collection of the Department of Food Science of the University of Udine (DIAL, Udine, Italy). The strain was maintained and grown on malt extract modified agar (Oxoid, Italy) supplemented with dextrose (1%) and peptone (1%) at 25 °C for 5–7 days. This specific strain was isolated from dry-cured ham (DCH) in a previous study and identified at the molecular level (accession number, KY608057.1). The *A. westerdijkiae* strain was grown on Czapek yeast extract agar plates (Oxoid, Italy) incubated at 25 °C for seven days. At the end of the incubation period, the conidia were removed from the culture surface according to the method reported by Virgili et al. [31] and suspended in peptonized water. Then, the conidial suspension was diluted to 10^4^, 10^6^, and 10^8^ conidia mL^−1^ using a hemocytometer. 

### 2.3. Inhibition of A. westerdijkiae by D. hansenii and L. buchneri Strains

The inhibition assay was performed using the method described by Virgili et al. [31] and Bleve et al. [40] and modified as follows: The experiments for *D. hansenii* were performed on MEMA adjusted to pH 6.0 and for *L. buchneri* on MRS (Oxoid, Italy). Top agar overlays were prepared by mixing *D. hansenii* suspensions in 10 mL of MEMA broth with 0.7% agar and 1%, 3%, and 5% NaCl (MEMA medium) and by mixing a *L. buchneri* suspensions in MRS broth with 0.7% agar, 0.3% yeast extract and 1%, 3%, and 5% NaCl added (MRSYE medium). Each of the top agars, containing different concentrations of *D. hansenii* or *L. buchneri* (10^2^, 10^4^, or 10^6^ CFU mL^−1^) prepared to investigate differences among different yeasts and LAB inocula, were transferred to Petri plates containing 15 mL of MEMA or MRS agar, respectively, resulting in a thick, continuous layer on the surface of the agars. Top agars without *D. hansenii* or *L. buchneri* were also spotted on MEMA and MRS as controls. Three 10 µL aliquots of *A. westerdijkiae* harboring 10^6^ conidia mL^−1^ were spotted on each top agar. The level of the *A. westerdijkiae* concentration was chosen to consider the worst degree of contamination. Three replicate experiments were performed for each yeast and LAB strain. Mold growth was expressed as the average measurement (mm) of two orthogonal diameters per colony after 14 days of incubation at 22 °C. The inhibitory activity was calculated using the equation reported by Lima et al. [41]:$$\%\;Inhibitory\;activity=\frac{\left(Mould\;grouth\;in\;control\;plate-Mould\;growth\;in\;treated\;plate\right)}{Mould\;growth\;in\;control\;plate}\times 100$$

### 2.4. Inhibitory Activity with Different Concentrations of D. hansenii and L. buchneri

*D. hansenii* (DIAL1) and *L. buchneri* (Lb4), which demonstrated the best performance in the aforementioned tests, were used. The method according to Virgili et al. [31] was used with the following modifications, and the concentration of yeast and LAB in the top agar was altered. Briefly, top agars (3% NaCl) containing different concentrations (10^2^, 10^4^, or 10^6^ CFU mL^−1^) of yeast, LAB and mixtures of both (Dh/Lb, at 1:1) ratio were used to examine the differences among yeasts and LAB at the different inocula concentrations. MRSYE medium was used to assess the activity of Dh/Lb. Ten milliliters of inoculated top agars were distributed onto 15 mL of MEMA or MRS medium, and three 10 µL suspensions containing 10^8^ conidia mL^−1^ of the mold were then spotted separately onto each plate and incubated at 22 °C for 14 days. Moreover, in this case the level of the *A. westerdijkiae* concentration was chosen to consider the worst degree of contamination. Three replicate experiments were performed for each test. Top agar without *D. hansenii* and *L. buchneri* and Dh/Lb inocula were also spotted as controls on MEMA, MRS, and MRSYE medium, respectively and inoculated by spots with *A. westerdijkiae.*

### 2.5. Inhibitory Activity at Different Concentrations of A. westerdijkiae

The abovementioned method [31] was used with the following modifications. The concentration of microorganisms in top agars (3% NaCl) was 10^6^ CFU mL^−1^. *D. hansenii* DIAL1, *L. buchneri* Lb4, and Dh/Lb, which exhibited the best performance in the aforementioned test, were used. Ten milliliters of inoculated top agar were distributed onto 15 mL of MEMA or MRS. MRSYE medium was used to assess the activity of Dh/Lb. Three 10 µL suspensions containing different concentrations of conidia (10^4^,10^6^, and 10^8^ conidia mL^−1^) of *A. westerdijkiae* were then separately spotted onto each plate and incubated at 22 °C for 14 days. Three replicate experiments were performed. Top agar without *D. hansenii* and *Lb. buchneri* and Dh/Lb inocula were also spotted as controls on MEMA, MRS, and MRSYE medium respectively and inoculated by spots with *A. westerdijkiae.*

### 2.6. Inhibitory Activity of D. hansenii and L. buchneri and Dh/Lb on OTA Production in the Dry-Cured Ham Model System 

The Inhibitory Activity of *D. hansenii* and *L. buchneri* and Dh/Lb on OTA Production in the Dry-Cured Ham Model System was performed using the method described by Battilani et al. [12], modified. Four SSDCH with different Aw values were collected from a San Daniele facility. From each SSDCH, samples (50 mm in diameter and 5 mm in height) were excised using a hollow metal sampler with a cylindrical cutting edge. The Aw values of the samples were measured with an AquaLab CX-2 instrument (Steroglass, Pullman, WA, USA) as 0.960 ± 0.005, 0.940 ± 0.005, 0.920 ± 0.005, and 0.900 ± 0.005. After being dipped in absolute ethanol for 2 min, the samples were removed and flamed to sterilize them prior to inoculation [7]. Then, the samples were placed in Petri plates, inoculated, and placed in sealed boxes equipped with beakers containing NaCl solutions [42] with the same Aw values as the four SSDCH samples. The salt solutions were prepared with distilled water (w/v) at 6.57% NaCl (Aw 0.96), 9.38% NaCl (Aw 0.94), 11.90% NaCl (Aw 0.92), and 14.18% NaCl (Aw 0.90). Each Aw value was confirmed using the AquaLab CX-2 instrument. The samples were incubated at 20 °C for 30 days in the dark, because usually great part of the dry cured ham seasoning is preferably made in the dark to eliminate the risk of fat photooxidation. Four different inocula or co-inocula, spotted on each samples, were tested: (1) Control *A. westerdijkiae* (10^4^ conidia cm^−2^); (2) *D. hansenii* DIAL1 (10^6^ CFU g^−1^) versus *A. westerdijkiae* (10^4^ conidia cm^−2^); (3) *L. buchneri* Lb4 (10^6^ CFU g^−1^) versus *A. westerdijkiae* (10^4^ conidia cm^−2^); and (4) a mix of *D. hansenii* DIAL1 + *L. buchneri* Lb4 (1/1 ratio; 10^6^ CFU g^−1^) versus *A. westerdijkiae* (10^4^ conidia cm^−2^). Each condition was evaluated in triplicate. At the end of the incubation period, the 48 samples were collected and analyzed for OTA according to the method reported by Matrella et al. [17]. Briefly: 10 g sample was homogenized with 6 mL of 1M phosphoric acid using an UltraTurrax T25 homogenizer for a 5 min. A 2.5 g aliquot of the homogenate was extracted with 5 mL of ethyl acetate and then centrifuged for 5 min at 4000 rpm. The organic phase was removed, re-extracted and reduced to approximately 3 mL and back-extracted with 3 mL of 0.5M NaHCO_3_, pH 8.4. The aqueous extract was acidified to pH = 2.5 with 7M H_3_PO_4_ and briefly sonicated to strip the CO_2_ formed. OTA was finally back extracted into 3 mL ethylacetate; the organic phase was evaporated to dryness under N_2_ stream, reconstituted in 150 µL of mobile phase and a 20 µL aliquot injected in the HPLC apparatus consisted of a Dionex P680 LPG pump. The fluorescence detector was a Varian mod. 9070; fluorescence excitation and emission wavelengths were 334 and 460 nm, respectively. 

### 2.7. Inhibitory Activity of Yeasts towards OTA Production during Dry-Cured Ham Seasoning

Sixteen fresh pork legs (FPLs) were collected. Each FPL (12 kg) was trimmed according to the traditional procedure and was then salted and treated until the period of first drying, at which time the FPLs were inoculated as follows. Four FPLs were inoculated, by spotting, with a suspension of *A. westerdijkiae* (final concentration of 10^4^ conidia/cm^2^); 4 FPLs were inoculated with a mix of *A. westerdijkiae* and *D. hansenii* (final concentration 10^4^ conidia/10^6^ CFU cm^−2^); 4 FPLs were inoculated with a mix of *A. westerdijkiae* and *L. buchneri* (10^4^ conidia/10^6^ CFU cm^−2^); and 4 FPLs were inoculated with a mix of *A. westerdijkiae* and *D. hansenii + L. buchneri* (10^4^ conidia/10^6^ CFU cm^−2^). *D. hansenii* DIAL1 and *L. buchneri* Lb4 strains were used. All of the FPLs were seasoned for six months according to the traditional procedure of SSDCH (Table 1). At the end of the second drying and seasoning period, 100 cm^2^ samples were taken from a depth of 0.5 cm below the meat surface and analyzed for OTA and for *D. hansenii*, *L. buchneri,* and *A. westerdijkiae* growth. Briefly, meat collected from a depth of 0.5 cm below the surface was homogenized using a Stomacher instrument (Lab Blender 400, PBI, Milan, Italy), and 10 g of the homogenate was then used for the analysis. Additional 10 g were sampled and used to value *D. hansenii*, *L. buchneri,* and *A. westerdijkiae* growth. *D. hansenii* was counted in malt extract modified agar (Oxoid, Italy) medium supplemented with dextrose (1%) and peptone (1%) and tetracycline chloride (10 mg L^−1^; Sigma, Milan, Italy) incubated at 25 °C for 3 days, the LAB in MRS agar (Oxoid, Italy) added with Delvocid (25 mg L^−1^, DSM, Food Specialities, Leewuarden, the Netherlands) incubated at 37 °C for 48-72 h, and *A. westerdijkiae* in Czapek yeast extract agar plates (Oxoid, Italy) added with tetracycline chloride (10 mg L^−1^; Sigma, Italy) incubated at 25 °C for seven days.

OTA was extracted and evaluated according to the method described by Matrella et al. [17]. Before sampling for OTA analysis, the surface of each SSDCH was also observed with a stereoscope (320X; WILD M 420, Heerbrugg, Switzerland) to assess the presence of mold growth (i.e., hyphae). In the case of presence of hyphae, they were isolated in MEMA and the grown colonies were identified comparing to the *A. westerdijkiae* ones. 

### 2.8. Sensorial Analysis

To evaluate the influence of the yeast and LAB culture starter on the organoleptic characteristics of the product, a sensory analysis was performed using the triangle test methodology UNI EN ISO 10399, triangle test (2018). FPLs were divided into 4 lots. Before starting the first drying step, lot A (3 FPL) was inoculated with a suspension of *D. hansenii* DIAL1 (10^6^ CFU cm^−2^), lot B (3 FPL) was inoculated with *L. buchneri* Lb4 (10^6^ CFU cm^−2^), lot C (3 FPL) was inoculated with Dh/Lb, and lot D (3 FPL) was not inoculated with either culture and represents the control. All FPLs were seasoned according to the traditional procedure (Table 1). At the end of the seasoning period (6 months), all the lots were subjected to the triangle test to compare their quality. Twenty non-professional assessors were presented with three products, two of which were identical. Non-professional assessors were chosen, because they represent the typical consumers, and were asked to state which product they believed was a unique sample. The assessors who indicated the presence of two distinct samples were presented the four different lots and asked to identify the best sample, considering the follow parameters compact and homogeneous slices; ruby red color; whitish fat; compact and not elastic consistency; delicate and distinctive bouquet; delicate taste; and no spices flavors perception. 

### 2.9. Statistical Analysis

The values of the various parameters were compared through one-way analysis of variance. The averages were compared with Tukey’s honest significance test using the Statistical Graphics software package (STSC, Inc. Rockville, MD, USA).

## 3. Results

Different inhibitory activities against *A. westerdijkiae* were observed among the tested strains in vitro (*p* < 0.05). *D. hansenii* DIAL1 and *L. buchneri* Lb4 showed the highest inhibitory activities against *A. westerdijkiae* for all the tested media and NaCl concentrations (Table 2 and Table 3). The diameters of *A. westerdijkiae* colonies were reduced by approximately 75%, 70%, and 72% by *D. hansenii* DIAL1 and by approximately 38.2%, 35.5%, and 25.5% by *L. buchneri* Lb4 in medium containing 1%, 3%, and 5% NaCl, respectively. Based on these results, both strains were selected to further assess their inhibitory effects in vitro and in situ. Different concentrations of both *D. hansenii* DIAL1 and *L. buchneri* Lb4 strains were tested separately and in combination to evaluate their activities against *A. westerdijkiae*. As expected, the inhibitory activity was dependent on both the tested strains. Indeed, in vitro concentrations of approximately 10^6^ CFU mL^−1^ of *D. hansenii* and *L. buchneri* showed a high inhibitory effect (*p* < 0.05; Table 4). 

The antagonistic activities of *D. hansenii* and *L. buchneri* and Dh/Lb (10^6^ CFU mL^−1^) were evaluated separately against different concentrations of *A. westerdijkiae* (10^2^, 10^4^, and 10^6^ conidia mL^−1^). The percentage of inhibition varied according to the mold concentration (Table 5), with the lowest inhibition observed after inoculation with the highest mold concentration (*p* < 0.05). The activity of *D. hansenii* alone was higher than that of *L. buchneri* but lower than the ones of Dh/Lb. At 10^6^ CFU mL^−1^, Dh/Lb completely inhibited *A. westerdijkiae* at 10^2^ and 10^4^ conidia mL^−1^ (data not shown), which was not observed when the same concentrations of *D. hansenii* and *L. buchneri* were used. These results demonstrated that the inhibitory activity of Dh/Lb was higher than that of *D. hansenii* and *L. buchneri* alone.

A reduction in mold growth limited OTA production as observed in SSDCH model systems. Table 6 displays the effects of *D. hansenii*, *L. buchneri,* and Dh/Lb on OTA production by *A. westerdijkiae* in SSDCH slices with different Aw levels. The results showed that the SSDCH samples inoculated with either strain alone or with Dh/Lb contained lower OTA concentrations than those inoculated with *A. westerdijkiae* alone (*p* < 0.05). Indeed, the OTA concentration in the SSDCH co-inoculated with *D. hansenii* and *A. westerdijkiae* was less than 2 µg kg^−1^ (Table 6). *L. buchneri* was less effective than *D. hansenii* and permitted *A. westerdijkiae* growth, which produced an OTA concentration over 10 µg kg^−1^. In contrast, OTA concentrations of 0.2–0.4 µg kg^−1^ were observed in SSDCH inoculated with mD/L, which is less than the limit (1 µg kg^−1^) allowed in meat and meat products by the Italian Ministry of Health (Circular no. 10-93 09/06/1999). As expected, slices of SSDCH with the higher Aw values (i.e., 0.96 and 0.94) exhibited the highest OTA concentrations (Table 6). 

The inhibitory activity of both microorganisms or of Dh/Lb against OTA-producing *A. westerdijkiae* during SSDCH production was assessed in situ (Table 7). Both microorganisms were co-inoculated separately or together (Dh/Lb) in FPLs with the *A. westerdijkiae* strain. All the inocula, were added at the beginning of the first drying phase. *D. hansenii* and *L. buchneri* grew till the end of the second drying phase, reaching 8.3 ± 1.2 and 6.8 ± 1.1 log CFU cm^−2^, respectively (*p* < 0.05). Additionally, the mix culture Dh/Lb grew, reaching 7.5 ± 0.8 and 6.5 ± 0.2 log CFU cm^−2^, respectively (Table 7). Then during the seasoning phase, the concentration of both the microorganisms decreased due to the Aw, which reached the value of 0.91. In particular the decrease for both the strains was between 0.3 and 0.5 log CFU cm^−2^. *D. hansenii* and the mix culture Dh/Lb blocked the growth of *A. westerdijkiae,* and its concentration remains at level of the inoculum (4.0 log CFU cm^−2^) and, consequently (*p* < 0.05), the concentration of OTA was at level of 1.0 ± 0.3 and < 1.0 μg kg^−1^, respectively (Table 7). In addition, also *A. westerdijkiae*, grew till the end of the seasoning phase, when inoculated alone or with *L. buchneri*, reaching 7.2 and 5.2 log CFU cm^−2^ (*p* < 0.05). The growth is also confirmed by the presence of OTA and by the presence of abundant hyphae on the FPLs, counted and identified in Czapek agar just as *A. westerdijkiae*. However, *L. buchneri* reduced the activity of *A. westerdijkiae,* which reached values about 4.8 log CFU cm^−2^ at the end of the second drying phase and 5.2 log CFU cm^−2^ at the end of seasoning phase, and consequently the values of OTA were at level of 5.5 ± 0.8 μg kg^−1^ and 13.2 ± 0.7 μg kg^−1^, respectively. Vice versa in samples where *A. westerdijkiae* was alone inoculated, its concentration reached values about 6.2 log CFU cm^−2^ at the end of the second drying phase and 7.2 log CFU cm^−2^, at the end of seasoning phase, presenting OTA values at level of 58.5 ± 4.5 μg kg^−1^ and 110.5 ± 0.6 μg kg^−1^, respectively (Table 7). 

Therefore, the data demonstrated that the Dh/Lb inoculum was able to reduce or completely inhibit the growth *A. westerdijkiae*, both in vitro and in situ. Based on these promising results, the use of starter cultures of *D. hansenii* and of mix of *D. hansenii* and *L. buchneri* to control OTA-producing molds may be recommended, considering that they could permit to respect the limit proposed by the Italian Ministry of Health (Circular no. 10-93 09/06/1999).

However, before using both strains in the starter, it needed to evaluate their influence on the sensorial acceptability of the product. At the end of seasoning period (Table 1), SSDCH samples underwent a triangular test by a panel of 20 non-professional assessors. The results demonstrated the acceptability of the SSDCH supplemented with the starter cultures and no difference among lots A, B, C (with bio-protective cultures), and D (without protective cultures) were perceived (Table 8). 

At sight, all the SSDCH slices, either inoculated or uninoculated, appeared compact and homogeneous. The color of the lean part was ruby red, whereas the fat part looked perfectly white, without any trace of oxidation (yellowish). Texture was compact, not elastic or chewy, the aromatic bouquet was mild and distinctive, as well as the taste was delicate, without any perception of spices or negative off-flavors. 

## 4. Discussion

The goal of this study was to evaluate the effect of the two selected microorganisms on the growth and OTA production of *A. westerdijkiae*. To this end, six selected meat product-native yeast and six LAB starter strains, four of which were isolated from meat products and two were from commercial starter cultures, were separately inoculated or co-inoculated with OTA-producing mold either in vitro or in situ. The strains of *Debaryomyces hansenii* and *Lactobacillus buchneri* were selected because they are widespread on meat and meat products [33,43,44] and are considered safe by the food industry [25]. SSDCHs are typical meat products of San Daniele and Parma areas of Italy. They are produced by the same pork raw meat, salt and process technology of San Daniele or Parma dry-cured ham, except for the seasoning time, which is shorter. Indeed, SSDCHs are seasoned for 6 months, while PDO products over 13 month and consequently they cannot acquire the brand PDO. During the drying and seasoning process, a large population of microorganisms (primarily molds and yeasts) develops on the outer layers of dry-cured hams [1,3,7,8,9,10,11,12,13]. Some of these molds can produce OTA, such as *P. nordicum*, *A. ochraceus,* and *A. westerdijkiae.* Under some environmental conditions, such as high R.H., these molds can grow and synthesize OTA, which can represent a real risk towards human health. Various studies have determined OTA levels in different meats and meat products contaminated by ochratoxigenic molds [1,7,12,13,45,46]. Different methods can be used to inhibit OTA-producing molds [1,13,47], the most common of which consists of spreading starter cultures on the meat product surface during the late drying and seasoning stages [1,28,29,30,31,36,47]. Mold, yeast and LAB starters are usually used in meat and food products because they limit the growth of pathogenic microorganisms by competing for nutrients and space and producing hydrolytic enzymes and toxins, and they also improve the sensorial quality of the product by secreting volatile compounds [1,36,48,49,50,51,52,53,54]. For yeasts and molds, the inhibitory mechanism is not fully understood, but it has been suggested that yeasts restrict nutrient availability and colonization sites, limiting the growth of OTA-producing molds [26]. However, it seems that the expression of non-ribosomal peptide synthetase gene linked to the OTA biosynthetic pathway of *P. verrucosum* is inhibited in presence of *D. hansenii* [32]. The present study was focused on assessing the use of *D. hansenii* and *L. buchneri* strains, both in vitro and in situ, against the OTA producer *A. westerdijkiae.* Under in vitro conditions, six *D. hansenii* and six *L. buchneri* strains significantly inhibited *A. westerdijkiae* growth. In particular *D. hansenii* DIAL1 and *L. buchneri* Lb4 showed the highest efficiencies on solid media. The inhibitory effect of *D. hansenii* DIAL1 was higher compared with that observed for *L. buchneri* Lb4 in all the in vitro tests, regardless to the salt concentration and the level of the *D. hansenii*, *L. buchneri,* and *A. westerdijkiae* inoculum. However, the inhibitory effects of Dh/Lb mix were higher than those obtained using the separately inoculated strains. The inhibitory activity of *D. hansenii* strains against *P. nordicum* and *A. ochraceus* is well known [1,31]. Recently, Iacumin et al. [1] demonstrated a high inhibition activity of *D. hansenii* vs *P. nordicum*; higher than the inhibitory effect of *D. hansenii* vs *A. westerdijkiae.*


In particular, different strains of *D. hansenii*, *Candida famata*, *Saccharomycopsis fibuligera*, *Candida zeylanoides,* and *Hyphopichia burtonii* have been selected for their ability to grow in dry-cured ham-like substrates and have been shown to be highly effective against *P. nordicum* and *A. ochraceus*. Indeed, the inhibitory effect was observed at the strain level [1,29]. The results obtained using the *D. hansenii* strain assayed in this study confirmed the previous data from above studies. In vitro, the efficiency depends on the strain, the medium and the Aw value and on the lysed yeast cells, which may provide nutrients for *P. nordicum*, *A. ochraceus,* or *A. westerdijkiae* growth [1,29]. The inhibitory effects could be due to compounds and antimycotic compounds produced by yeast and LAB strains [1,29,49,55] and mostly to the competition for nutrients [1,36,38,56]. Due to the large variability in OTA-producing molds on dry-cured ham and meat products [1,20,31,34,36,57], yeasts and LAB inoculated at low concentrations were shown to be associated with low levels of mold inhibition [1]. In this study, the use of different yeast and LAB concentrations in vitro resulted in different *A. westerdijkiae* inhibitory effects, demonstrating that mold inhibition could depend on the competition for nutrients and on antimycotic factor production. Consequently, the highest yeast and LAB concentration (10^6^ CFU g^−1^) assayed induced the largest inhibitory effect against the OTA-producing molds. Indeed, it was observed that the lower inoculum of protective culture produced either smaller quantities of antimould compounds or less competition for nutrients, and, consequently, the inhibition of *A. westerdijkiae* has been lower. The results confirmed the potential biocontrol effect of both yeast and LAB strains used in this study. Our data agree with the results reported by Virgili et al. [31] and Metfah et al. [36]. Furthermore, the inhibitory activity of the yeast and LAB was affected by the mold concentration. Virgili et al. [31] hypothesized that the concentration of OTA-producing molds in meat products is a key factor for the effectiveness of yeast starters. The use of LAB and a mix of yeast and LAB starter must be considered and evaluated. In this study, the use of bioprotective agents, including a mix of LAB and yeast, produced the best results in reducing the growth or OTA production of *A. westerdijkiae.* Furthermore, in the SSDCH model system, the data demonstrated the potential for the effective biocontrol of the OTA-producing *A. westerdijkiae* strain by *D. hansenii* DIAL1 and *L. buchneri* Lb4. The results appear to be strictly dependent on the Aw value and the bioprotective strains added separately or in combination. The reduced OTA concentration in the SSDCH model system with different Aw values compared with that observed in the control samples was rather notable. The antagonistic activities of *D. hansenii* and *L. buchneri* were also dependent on the Aw value and were different from the data reported by Simoncini et al. [30], who demonstrated that the variability in the antagonistic effect was affected by the strains of the different species used (*D. hansenii* and *C. zeylanoides*). In our study, the Aw value probably influenced either the bioprotective strains or *A. westerdijkiae* growth, as demonstrated by the control plate results, where the bioprotective strains were not added and the decreasing OTA concentration was correlated to the decrease in the Aw value. Moreover, in the SSDCH model system the OTA-reducing effects of *D. hansenii* were higher than those of *L. buchneri*, but the greatest effects were observed using a mix of both bioprotective strains. The bioprotective strains, alone or in combination, had positive effects on the inhibition of mold growth and OTA production and accumulation. The level of OTA presence in SDDCH was not always correlated to the level of *A. westerdijkiae* concentration. Indeed, no association was observed between the presence of mycotoxin and the biomass of the *A. westerdijkiae* OTA-producing strain, confirming the results of Xu et al. [58], who demonstrated that the production and concentration of some mycotoxins in food are not necessarily proportional to the biomass of the OTA-producing mold. Again, the data confirmed the biocontrol activity of *D. hansenii* against OTA-producing mold in meat products as previously demonstrated by various authors [1,26,29,31,52]. However, there are no data on the biocontrol activity of *L. buchneri* against OTA-producing molds, even though various LAB species isolated from different raw and processed foods can inhibit either the growth of mycotoxigenic molds or reduce the presence of mycotoxins [30,31,40,56]. Thus, this study provides the first demonstration of the use of an *L. buchneri* strain for the biocontrol of OTA-producing mold. OTA was only detected at high levels (58.5 ± 4.5 μg kg^−1^ at the end of the second drying phase and 110.5 ± 0.6 μg kg^−1^ at the end of seasoning phase) in the SSDCH produced without the addition of the protective cultures. Additionally, in situ the best inhibition of *A. westerdijkiae* was obtained using the Dh/Lb inoculum. Indeed, in the FPLs individually inoculated with the bioprotective strains, *A. westerdijkiae* grew and produced OTA at a higher level than the limit proposed by the Italian Ministry of Health (1 μg kg^−1^, Circolare Ministero Sanità No. 30 10-09/06/1999). OTA production was detected in the meat in the area where *A. westerdijkiae* grew, producing a yellow slime that covered the entire meat portion of the FPLs, but no growth was observed on the skin. The inoculation of the bioprotective strains either separately or in combination, with *A. westerdijkiae* reduced or did not permit its growth on the FPLs. Therefore, the dominance of the bioprotective strains was the primary parameter involved in the elimination of *A. westerdijkiae* growth and consequently the elimination of the OTA presence and production. Indeed, at the end of the seasoning phase, the amount of OTA present was less than the limit proposed by the Italian Ministry of Health (Circolare Ministero Sanità No. 30 10-09/06/1999). In a previous study [1], *D. hansenii* was shown to inhibit *A. ochraceus* or *P. nordicum* in speck and, consequently, eliminated the presence of OTA. Furthermore, several studies [29,30,31] have demonstrated that inoculated and native *D. hansenii, C. zeylanoides* and *H. burtonii* were able to dominate *A. ochraceus* and OTA-producing molds, showing the occurrence of an antagonistic variability at the level of the tested strains. However, *D. hasenii* DIAL 1 produced better inhibition versus the OTA producer *A. ochraceus* in speck [1] than that observed versus *A. westerdijkiae* in SSDCH. Regarding the decreased activity of *D. hansenii* DIAL1 against *A. westerdijkiae*, it should be noted that the latter strain is more adapted to temperate climates, exhibiting abundant growth at 22 °C, and it typically produces OTA in higher amounts and more consistently than *A. ochraceus* [36,59]. However, although the Aw value typically has a significant effect on OTA levels, decreasing the Aw did not increase the concentration of OTA produced by *A. westerdijkiae* in the SSDCH model system, since OTA biosynthesis is naturally activated under sub-optimal, weak stress conditions such as 3%–5% NaCl [60]. Therefore, it was necessary to select the strain with the highest antagonistic effect among the isolated species. This finding was confirmed in this study and in a previous study [1]. The significant reduction of OTA appeared to be related to the inhibition of *A. westerdijkiae* growth by the co-inoculated yeasts and LAB. No other hypotheses could be formulated, despite the observation that environmental conditions have different effects on *A. westerdijkiae* OTA production and growth in a sterile dry-cured ham-based medium and that OTA production does not correlate with growth [61]. Furthermore, no mold hyphae were observed on the surfaces of the co-inoculated SSDCH model system and in SSDCH under a stereomicroscope. Consequently, the possibility that the great reduction in OTA levels in the samples treated with a combination of both bioprotective starters may be due to mycotoxin degradation by the yeasts into less toxic compounds was also excluded [1,29,30]. Again, the possibility that OTA could be adsorbed on the yeast cell wall was excluded, despite this activity having been shown for *S. cerevisiae* and *D. hansenii* [62,63]. Although the mechanisms by which the mixed bioprotective strains reduce OTA levels need to be elucidated [29,64], the stereoscope observations demonstrated that the reduced OTA levels in the SSDCH model system and SSDCH samples was strictly due to the dominating effect of the mixed bioprotective strains over the OTA-producing *A. westerdijkiae*.

Despite the antagonistic effect observed in the SSDCH model system at all of the Aw levels tested, it is reasonable to propose that the starter culture should be inoculated before the first drying phase, when the Aw is 0.95 ± 0.01 and *A. westerdijkiae* has not started to grow. The suggested concentration of the mixed inoculum should be 10^6^ CFU cm^−2^ to absolutely ensure that the inoculum can predominate over the *A. westerdijkiae*. The results suggested that adding this concentration of the mixed starter at the specific production steps confirms the results of different studies [1,13,18,29,30,31,32] in which the addition of the starter at the beginning of the production process was suggested for dry-cured hams, when the Aw of the product is still high (0.94–0.95) and can support OTA-producing mold growth. In our study, the mixed starter culture grew rapidly on the SSDCH at this Aw value and blocked the growth of OTA-producing *A. westerdijkiae*. Therefore, the biopreservative effect of the tested strains was obvious. Consequently, the combination of both strains may be proposed as antagonistic agents to prevent the presence of OTA-producing molds and the bioaccumulation of OTA during SSDCH production (Figure 1). 

Finally, as suggested in a previous study [1] for speck production, the application of bioprotective using an adequate hygienic system based on hazard analysis and critical control points (HACCP) can absolutely prevent the presence of OTA in SSDCH. The HACCP system can reduce the level of OTA-producing mold contamination on FBCs and SSDCH and flavor the antagonistic effect of the bioprotective starter [1]. Additionally, the raw material, temperature, and R.H. of the drying and seasoning rooms should be controlled [57]. 

The sensorial acceptability of all the SSDCHs, either inoculated or uninoculated with the bioprotective starter, was confirmed by a triangular test using a panel of 20 non-professional assessors. These assessors did not observe any difference between lots A, B, and C (with bioprotective cultures) and lot D (uninoculated control). 

## 5. Conclusions

*D. hansenii* and *L. buchneri* are potential biopreservative agents for use in eliminating the growth of the OTA producer *A. westerdijkiae* in SSDCH, a recent Italian dry-cured ham product. The use of selected *D. hansenii* DIAL1 and *L. buchneri* Lb4 starter cultures, the control of raw meat and the technological (temperature and R.H.) and hygienic parameters are fundamental for the reduction of health hazards due to the growth of the OTA producer *A. westerdijkiae* in SSDCH. Consequently, the inoculation of *D. hansenii* or *L. buchneri* as mixed starter culture strains after the first drying stage could improve the safety and quality of SSDCH.

## Figures and Tables

**Figure 1 microorganisms-08-01623-f001:**
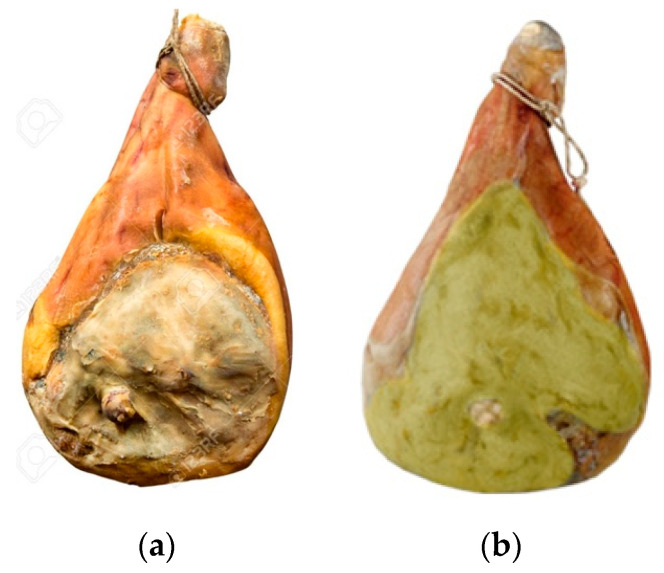
Short-seasoned dry-cured ham (SSDCH) (**a**) without; (**b**) with *A. westerdijkiae.*

**Table 1 microorganisms-08-01623-t001:** Flowchart of dry cured ham production.

Production Phases	Time	Temperature	R.H. %
Raw meat-Trimming	24 h	1–7 °C	50–60
Brining	15 day	1–4 °C	75–88
Pressing	2 day	4 °C	70–75
Rest	50 days	4–10 °C	70–75
Washing	5 h		
I Drying	7 days	20–21 °C	90
II Drying/pre-seasoned	20 day	14–20 °C	70–85
Seasoning	3 months	15–22 °C	70–80

R.H.: Relative Humidity.

**Table 2 microorganisms-08-01623-t002:** Percentage of inhibitory activity of *Debaryomyces hansenii* vs *A. westerdijkiae* co-inoculated in agar plates.

Starter Yeast	% Inhibitory on *A. westerdijkiae*		
	1% NaCl	3% NaCl	5% NaCl
*D. hansenii* DIAL1	75.0 ± 0.5a	70.0 ± 0.3a	72.0 ± 0.1a
*D. hansenii* DIAL2	68.3 ± 0.2b	67.2 ± 0.8b	70.2 ± 0.3b
*D. hansenii* DIAL3	70.3 ± 0.5b	68.5 ± 0.2c	68.3 ± 0.3c
*D. hansenii* DIAL4	70.0 ± 0.2b	69.3 ± 0.2c	69.3 ± 0.7c
*D. hansenii* DIAL5	68.5 ± 0.2b	69.0 ± 0.2c	68.0 ± 0.3c
*D. hansenii* DIAL6	70.3 ± 0.5b	69.3 ± 0.2c	66.5 ± 0.3d

Data represent the means ± standard deviations of the total samples; mean with the same letters within the same column (following the values) are not significantly differently (*p* < 0.05).

**Table 3 microorganisms-08-01623-t003:** Percentage of inhibitory activity of *Lactobacillus buchneri* vs *A. westerdijkiae* co-inoculated in agar plates.

Starter LAB	% Inhibitory on *A. westerdijkiae*		
	1% NaCl	3% NaCl	5%NaCl
*L. buchneri* LABb2	20.0 ± 0.2c	15.0 ± 0.2c	10.0 ± 0.3b
*L. buchneri* LABb3	23.0 ± 0.3b	17.0 ± 0.3b	12.0 ± 0.5c
*L. buchneri* Lb1	16.5 ± 0.3a	11.3 ± 0.5a	9.5 ± 0.2a
*L. buchneri* Lb2	22.3± 0.3b	18.2 ± 0.5d	12.2 ± 0.5c
*L. buchneri* Lb3	24.0 ± 0.5d	21.3 ± 0.4e	18.5 ± 0.2d
*L. buchneri* Lb4	38.2 ± 0.2e	35.5 ± 0.4f	25.5 ± 0.2e

Data represent the means ± standard deviations of the total samples; mean with the same letters within the same column (following the values) are not significantly differently (*p* < 0.05).

**Table 4 microorganisms-08-01623-t004:** Percentage of inhibitory effect of different concentration of *D. hansenii* and *L. buchneri* vs *A. westerdijkiae* co-inoculated in agar plates.

Microorganisms	Inhibitory Effect on *A. westerdijkiae*
***D. hansenii* DIAL1**	
10^2^ CFU mL^−1^	31.0 ± 0.2a
10^4^ CFU mL^−1^	50.4 ± 0.3b
10^6^ CFU mL^−1^	74.8 ± 0.4c
***L. buchneri* Lb4**	
10^2^ CFU mL^−1^	19.0 ± 0.4a
10^4^ CFU mL^−1^	21.2 ± 0.3b
10^6^ CFU mL^−1^	38.2 ± 0.2c
**Dh/Lb**	
10^2^ CFU mL^−1^	43.5 ± 0.5a
10^4^ CFU mL^−1^	68.0 ± 0.4b
10^6^ CFU mL^−1^	95.0 ± 0.1c

Data represent the means ± standard deviations of the total samples; mean with the same letters within the same column (following the values) are not significantly differently (*p* < 0.05). *A. westerdijkiae* 10^6^ conidia mL^−1^; n.d.: Not done. Dh/*L. buchneri* (Lb): Mix *D. hansenii* and *L. buchneri.*

**Table 5 microorganisms-08-01623-t005:** Percentage of inhibitory effect of *D. hansenii* and *L. buchneri* vs different concentration of *A. westerdijkiae* co-inoculated in agar plates.

*A. westerdijkiae*	Inhibitory Effect of		
	*D. hansenii*	*L. buchneri*	Dh/Lb
10^2^ CFU mL^−1^	65.0 ± 0.3a	38.0 ± 0.6a	75.0 ± 0.2a
10^4^ CFU mL^−1^	58.4 ± 0.7b	30.2 ± 0.3b	66.5 ± 0.2b
10^6^ CFU mL^−1^	45.0 ± 0.5c	8.0 ± 0.5c	55.0 ± 0.3c

Data represent the means ± standard deviations of the total samples; mean with the same letters within the same column (following the values) are not significantly differently (*p* < 0.05). Dh/Lb: Mix *D. hansenii* and *L. buchneri; D. hansenii, L. buchneri,* and Dh/Lb. concentration: 10^4^ CFU mL^−1^.

**Table 6 microorganisms-08-01623-t006:** Mean production of Ochratoxin A by *A. westerdijkiae* in control and selected *D. hansenii* and *L. buchneri* co-inoculated in dry cured ham model system with different water activity (Aw).

Ochratoxin A (μg kg^−1^) in Dry-Cured Ham Model				
Aw	*A. westerdijkiae* Control	*A. westerdijkiae* vs. *D. hansenii*	*A. westerdijkiae* vs. *L. buchneri*	*A. westerdijkiae* vs. Dh/Lb
0.96	15.5± 0.2a	1.9± 0.2b	13.0± 0.4c	0.2 ± 0.1d
0.94	12.3± 0.3a	1.8± 0.2b	11.7± 0.3c	0.4 ± 0.2d
0.92	8.5± 0.2a	1.7± 0.1b	6.4± 0.2c	0.3 ± 0.1d
0.90	6.0± 0.5a	1.5± 0.2b	4.3± 0.4c	0.3 ± 0.1d

Data represent the means ± standard deviations of the total samples; mean with the same letters within the same lines (following the values) are not significantly differently (*p* < 0.05). Dh/Lb mix *D. hansenii* and *L. buchneri; D. hansenii, L. buchneri,* and Dh/Lb. concentration: 10^6^ CFU cm^−2^; *A. westerdijkiae* 10^4^ conidia cm^−2^.

**Table 7 microorganisms-08-01623-t007:** Mean production of Ochratoxin A by *A. westerdijkiae* in control and selected *D. hansenii* and *L. buchneri* co-inoculated at the end of second drying and seasoning stages and their fate during the seasoning of short seasoning dry cured ham.

Phase		OTA	*D. hansenii*	*L. buchneri*	*A. westerdijkiae*
		(mg kg^−1^)	(CFU cm^−2^)		
**End II drying**	***A. westerdijkiae* vs.**				
	*D. hansenii*	<1.0 a	8.3 ± 1.2a	n.i.	4.0 ± 0.1a
	*L. buchneri*	5.5 ± 0.8b	n.i.	6.8 ± 1.1a	4.8 ± 0.8b
	Dh/Lb	<0.1c	7.5 ± 0.8a	6.5 ± 0.2a	4.0 ± 0.3a
	***A. westerdijkiae***	58.5 ± 4.5d	n.i	n.i	6.2 ± 1.2c
**End seasoning**	*A. westerdijkiae* vs				
	*D. hansenii*	1.0 ± 0.3a	8.0 ± 0.5a	n.i.	4.0 ± 0.1a
	*L. buchneri*	13.2 ± 0.7b	n.i	6.0 ± 0.2a	5.2 ± 0.8b
	Dh/Lb	<0.1c	7.0 ± 0.3b	5.9 ± 0.5a	4.0 ± 0.3a
	*A. westerdijkiae*	110.5 ± 0.6d	n.i.	n.i.	7.2 ± 0.8c

Data: OTA mean ± standard deviations of 20 replicates: μg kg^−1^.; LOD < 0.1 μg kg^−1^; samples taken from a depth of 0.5 cm below the surface; Dh/Lb: Mix *D. hansenii* and *L. buchneri. D. hansenii, L. buchneri,* and Dh/Lb concentration: 10^6^ CFU cm^−2^; *A. westerdijkiae* 10^4^ conidia cm^−2^. n.i: Not inoculated. Data represent the means ± standard deviations of the total samples; inside the end second ripening and the end seasoning, mean with the same letters within the same columns (following the values) are not significantly differently (*p* < 0.05).

**Table 8 microorganisms-08-01623-t008:** Sensorial evaluation by not trained panelists.

Lot of Production	Difference	Final Values Scores *
Lot A vs. Lot B	- 20/20	2/2
Lot A vs. Lot C	- 20/20	2/2
Lot A vs. Lot D	- 20/20	2/2
Lot B vs. Lot C	- 20/20	2/2
Lot B vs. Lot D	- 20/20	2/2

-: No difference; lot A (3 FPL) was inoculated with a suspension of *D. hansenii* DIAL1 (10^6^ CFU cm^−2^), lot B (3 FPL) was inoculated with *L. buchneri* Lb4 (10^6^ CFU cm^−2^), and lot C (3 FPL) was inoculated with Dh/Lb and lot D (3 FPL) was not inoculated with either culture and represents the control. * Scores (Lot vs Lot) 1 (excellent), 2 (good), 3 (sufficient), and 4 (poor).

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
