# Peer review of "Effect of a Debaryomyces hansenii and Lactobacillus buchneri Starter Culture on Aspergillus westerdijkiae Ochratoxin A Production and Growth during the Manufacture of Short Seasoned Dry-Cured Ham"

_microorganisms, 2020, doi:10.3390/microorganisms8101623_

Round 1
Reviewer 1 Report
The work is a valuable contribution to the development of the science of bioprotection used to improve the microbiological quality of Italian hams. However, I would like the authors to make a few clarifications about the text.
Line 19-23: "The product is obtained using porklegs, which are saltedand seasoned for sixmonths. The ingredients are similar to Protected Denomination of Origin (PDO) products such as San Daniele or Parma dry-cured hams, which consist ofpork leg, salt and spices, but they are only seasoned for six months."
It follows from the above record that both types of ham are seasoned for six months. What is the seasoning time for a new type of ham and what is the seasoning time for traditional hams?
Lines 227-228: "The samples were incubated in the dark." What does this result from? Is this the standard procedure for making this type of ham?
Line 283-284: What is the reason for the commitment to sensory assessment of non-professional assessors instead of professionals? This can be a risky matter, because people without preparation in the sensory assessment often do not perceive subtle differences in the taste of the assessed products.
Lines 299-305: these sentences are redundant because all the above information has been described in detail in the methodological part of the article.
I also ask the Authors to read the text carefully before it is finally submitted for possible printing in terms of minor punctuation and editorial errors (Latin names of microorganisms should be written in italics - see Abstract).
Author Response
Answer to Reviwer 1:
The work is a valuable contribution to the development of the science of bioprotection used to improve the microbiological quality of Italian hams. However, I would like the authors to make a few clarifications about the text.
Line 19-23: "The product is obtained using porklegs, which are salted and seasoned for six months. The ingredients are similar to Protected Denomination of Origin (PDO) products such as San Daniele or Parma dry-cured hams, which consist ofpork leg, salt and spices, but they are only seasoned for six months."
Answer: Thanks I explain better: Lines 19-25 Recently, specific dry-cured hams have started to be produced in San Daniele and Parma areas. The ingredients are similar to Protected Denomination of Origin (PDO) produced in San Daniele or Parma areas, and include pork leg, coming from pigs bred in the Italian peninsula, salt and spices. However these specific new products can not be marked as PDO either San Daniele or Parma dry cured ham, because they are seasoned for 6 months, and the mark PDO is given only to products seasoned over 13 months. Consequently these products are called short-seasoned dry-cured ham (SSDCH) and are not branded PDO.
Lines 436-439 They are produced by the same pork raw meat, salt and process technology of San Daniele or Parma dry-cured ham, except for the seasoning time, which is shorter. Indeed SSDCHs are seasoned for 6 months, while PDO products over 13 month and consequently they cannot acquire the brand PDO.
It follows from the above record that both types of ham are seasoned for six months. What is the seasoning time for a new type of ham and what is the seasoning time for traditional hams?
Answer: Thanks as above reported : Lines 22-24 - the mark PDO is given only to products seasoned over 13 months.
Lines 227-228: "The samples were incubated in the dark." What does this result from? Is this the standard procedure for making this type of ham?
Answer: Thanks I explained better: Yes, this is the standard procedure for this type of ham. The use of the dark for the inhibitory activity of D. hansenii and L. buchneri and Dh/Lb on OTA production in the dry-cured ham model system, is due to eliminate problems related to fat and meat photooxidation which could interfere with the activity of the microorganisms used.
Lines 221-223 I corrected: The samples were incubated at 20 °C for 30 days in the dark, because usually great part of the dry cured ham seasoning is preferably made in the dark to eliminate the risk of fat photooxidation.
Line 283-284: What is the reason for the commitment to sensory assessment of non-professional assessors instead of professionals? This can be a risky matter, because people without preparation in the sensory assessment often do not perceive subtle differences in the taste of the assessed products.
Answer: Thanks – The choice of not professional assessors depends on the assessment made by classical consumers, which are not trained and accept the product only by their own perception and their response is good or not good.
I corrected: Lines 279-280 - It was chosen non-professional assessors, because they represent the typical consumers, and were asked to state which product they believed was a unique sample.
Lines 299-305: these sentences are redundant because all the above information has been described in detail in the methodological part of the article.
Answer: Thanks I eliminate these sentences
I also ask the Authors to read the text carefully before it is finally submitted for possible printing in terms of minor punctuation and editorial errors (Latin names of microorganisms should be written in italics - see Abstract).
Answer- Thanks I corrected all.

Reviewer 2 Report
The manuscript evaluates the effectiveness of L. buchneri and D. hansenii to counteract the presence of OTA in a specific type of Italian dry-cured ham. It is widely recognized that biocontrol is the most appropriate strategy to reduce the hazard associated with OTA in this product because of its specific properties. Although this is the topic of several published manuscripts, most of them have been based on P. nordicum, which states the novelty of the work. Additionally, the obtained results agree with those previously reported for D. hansenii. Despite the interest of the manuscript, several downsides have been detected, which are detailed below.
Please revise the use of abbreviations since many mistakes have been found throughout the manuscript. There are mistakes in the microbial species but also in other words. The English grammar must be also checked considering among other things that American or British English has to be used but not both. There are some mistakes in the units used for expressing concentration or volumes too.
When talking only about moulds “fungal” must be avoid since yeasts are also tested in the manuscript.
Regarding the Material and Methods section, the information given in lines 159-160 is a repetition of that included in lines 154-155. Besides, it is necessary to justify why different inocula of the mould were used in the assays described in sections 2.3. and 2.4. How the normality of data was tested is also missed.
In line 369, please check mD/L.
In the Results section, the data from the statistical analysis are missed in Table 7. Furthermore, it would be desirable to organize the results from the sensorial analysis in a table or in a figure including the applied statistical analysis. Even, some pictures from the final products would be acceptable. The discussion about such evaluation is also poor.
Minor mistakes have been found in the list of references.
Author Response
Answer to Reviewer 2:
The manuscript evaluates the effectiveness of L. buchneri and D. hansenii to counteract the presence of OTA in a specific type of Italian dry-cured ham. It is widely recognized that biocontrol is the most appropriate strategy to reduce the hazard associated with OTA in this product because of its specific properties. Although this is the topic of several published manuscripts, most of them have been based on P. nordicum, which states the novelty of the work. Additionally, the obtained results agree with those previously reported for D. hansenii. Despite the interest of the manuscript, several downsides have been detected, which are detailed below.
Please revise the use of abbreviations since many mistakes have been found throughout the manuscript. There are mistakes in the microbial species but also in other words. The English grammar must be also checked considering among other things that American or British English has to be used but not both. There are some mistakes in the units used for expressing concentration or volumes too.
Answer: Thanks I corrected them
When talking only about moulds “fungal” must be avoid since yeasts are also tested in the manuscript.
Answer: Lines 175-181 - Thanks I corrected it using mould.
Regarding the Material and Methods section, the information given in lines 159-160 is a repetition of that included in lines 154-155.
Answer : Thanks I eliminated the repetition.
Besides, it is necessary to justify why different inocula of the mould were used in the assays described in sections 2.3. and 2.4. How the normality of data was tested is also missed.
Answer: Thanks I added Lines 174-175 The level of the Aspergillus westerdijkiae concentration was chosen to consider the worst degree of contamination.
Line 193-195 - Also in this case the level of the Aspergillus westerdijkiae concentration was chosen to consider the worst degree of contamination.
In line 369, please check mD/L.
Answer: Thanks line 370, I corrected it.
In the Results section, the data from the statistical analysis are missed in Table 7.
Answer: Thanks I added the statistical analysis. Lines 393-402.
Table 7: Mean production of Ochratoxin A by A. westerdijkiae in control and selected D. hansenii and L. buchnerico-inoculated at the end of II drying and seasoning stages and their fate during the seasoning of short seasoning dry cured ham
|
Phase |
|
OTA |
D.hansenii |
Lb. buchneri |
A. westerdijkiae |
|
|||||||||||
|
|
|
mg kg-1 |
CFU cm-2 |
|
|||||||||||||
|
End II drying |
A. westerdijkiae versus |
|
|
|
|
|
|||||||||||
|
|
D. hansenii |
< 1.0 a |
8.3 ± 1.2a |
n.i. |
4.0 ± 0.1a |
||||||||||||
|
Lb. buchneri |
5.5 ± 0.8b |
n.i. |
6.8 ± 1.1a |
4.8 ± 0.8b |
|||||||||||||
|
Dh/Lb |
< 0.1c |
7.5 ± 0.8a |
6.5 ± 0.2a |
4.0 ± 0.3a |
|||||||||||||
|
A. westerdijkiae |
58.5 ± 4.5d |
n.i |
n.i |
6.2 ± 1.2c |
|||||||||||||
|
End seasoning |
A. westerdijkiae versus |
|
|
|
|
||||||||||||
|
D. hansenii |
1.0 ± 0.3a |
8.0 ± 0.5a |
n.i. |
4.0 ± 0.1a |
|||||||||||||
|
Lb. buchneri |
13.2 ± 0.7b |
n.i |
6.0 ± 0.2a |
5.2 ± 0.8b |
|||||||||||||
|
Dh/Lb |
< 0.1c |
7.0 ± 0.3b |
5.9 ± 0.5a |
4.0 ± 0.3a |
|||||||||||||
|
A. westerdijkiae |
110.5 ± 0.6d |
n.i. |
n.i. |
7.2 ± 0.8c |
|||||||||||||
Data: OTA mean ± standard deviations of 20 replicates: μg kg-1.; LOD < 0.1 μg kg-1; Samples taken from a depth of 0.5 cm below the surface; Dh/Lb: mix D. hansenii and L. buchneri. D. hansenii, L. buchneri and Dh/Lb.concentration: 106 CFU cm-2; A. westerdijkiae 104 conidia cm-2. n.i: not inoculated. Data represent the means ± standard deviations of the total samples; Inside the end II ripening and the End seasoning, Mmean with the same letters within the same columns (following the values) are not significantly differently (p < 0.05).
Furthermore, it would be desirable to organize the results from the sensorial analysis in a table or in a figure including the applied statistical analysis. Even, some pictures from the final products would be acceptable. The discussion about such evaluation is also poor.
Answer: Thanks I added the follow table demonstrating no difference about the sensory evaluation – Lines 4121-426.
At sight, all the SSDCH slices either inoculated or uninoculated appeared compact and homogeneous. The colour of the lean part was ruby red, whereas the fat part looked perfectly white, without any trace of oxidation (yellowish). Texture was compact, not elastic or chewy, the aromatic bouquet was mild and distinctive, as well as the taste was delicate, without any perception of spices or negative off-flavours.
Table 8: Sensorial evaluation by not trained panellists
|
Lot of production |
Difference |
Final values Scores* |
|
Lot A versus Lot B |
- 20/20 |
2/2 |
|
Lot A versus Lot C |
- 20/20 |
2/2 |
|
Lot A versus Lot D |
- 20/20 |
2/2 |
|
Lot B versus Lot C |
- 20/20 |
2/2 |
|
Lot B versus Lot D |
- 20/20 |
2/2 |
|
Lot C versus Lot D |
- 20/20 |
2/2 |
Legend: -: no difference; + n. positive assessments/total assessments; Lot A (3 FPL) was inoculated with
a suspension of D. hansenii DIAL1 (106 CFU cm-2), lot B (3 FPL) was inoculated with L. buchneri Lb4
(106 CFU cm-2), lot C (3 FPL) was inoculated with Dh/Lb and lot D (3 FPL) was not inoculated with either
culture and represents the control. *Scores (Lot versus Lot) 1 (excellent). 2 (good). 3 (sufficient). 4 (scarse).
Minor mistakes have been found in the list of references.
Answer Thanks I corrected all the mistakes.

Reviewer 3 Report
The submitted manuscript is original and it does expand the knowledge in the area of potential bio preservative agents for use in eliminating the growth of the OTA producing moulds in meat products (hams). I believe it significantly build on (the author's) previous work and fits the scope of the journal. The methodology presented in the manuscript and analysis provided are accurate and properly conducted. The writing style is clear and appropriate. Tables and figures are clear to read and labelled appropriately. However, there are some minor issues to be addressed:
#1: To the best of our knowledge PDO stands for Protected designation of origin as the name of a geographical region or specific area that is recognized by official rules to produce certain foods with special characteristics related to location. The PDO regulation covers agricultural products and foodstuffs that are produced, processed, and prepared in a given geographical area using recognized know-how in this specific zone. Therefore, it is very important for food producers and regulatory institutions to determine and quantify the specific quality parameters of such products to avoid fraud and to confirm their geographical origin. Therefore, we would like to ask the authors does Protected DENOMINATION of Origin is the same as Protected DESIGNATION of origin. If so, please explain how „a particular type of dry-cured ham has begun to be produced in DIFFERENT Italian areas“ can be branded as San Daniele or Parma dry-cured hams. If not, please explain what Protected DENOMINATION of Origin stands for. To be clear, we just want to point out that seasoning for 13 months is just one of the requirements for Parma dry-cured hams. From what we can understand from this manuscript, if SSDCH hams produced in DIFFERENT Italian areas are to be seasoned for 13 months, they could be branded as San Daniele or Parma dry-cured hams.
#2: As authors have rightfully acknowledged, inadequate processing procedures can increase the development of moulds that can lead to negative effects, such as the production of mycotoxins (OTA-producing moulds or fungus). Then the authors also state that the use of short seasoning times can cause high moisture levels on the surface of the hams, which has detrimental effects, such as the growth of OTA-producing moulds. Our conclusion is that short seasoning times can be considered as “inadequate processing procedure”. Perhaps we should advocate that this economy (money) driven practise (shortening of time) leads to unsafe products, instead of finding the ways to use various agents (chemical, physical and biological) for food detoxification purposes or limitation and inhibition of OTA-producing moulds. More so because the authors also stated that HACCP system can reduce the level of OTA-producing mould contamination on SSDCH.
#3: Line 33: LAB species in red font ?!
Therefore, I would recommend the paper to be published after minor revisions.
Author Response
Answer to Reviewer 3:
The submitted manuscript is original and it does expand the knowledge in the area of potential bio preservative agents for use in eliminating the growth of the OTA producing moulds in meat products (hams). I believe it significantly build on (the author's) previous work and fits the scope of the journal. The methodology presented in the manuscript and analysis provided are accurate and properly conducted. The writing style is clear and appropriate. Tables and figures are clear to read and labelled appropriately. However, there are some minor issues to be addressed:
#1: To the best of our knowledge PDO stands for Protected designation of origin as the name of a geographical region or specific area that is recognized by official rules to produce certain foods with special characteristics related to location. The PDO regulation covers agricultural products and foodstuffs that are produced, processed, and prepared in a given geographical area using recognized know-how in this specific zone. Therefore, it is very important for food producers and regulatory institutions to determine and quantify the specific quality parameters of such products to avoid fraud and to confirm their geographical origin. Therefore, we would like to ask the authors does Protected DENOMINATION of Origin is the same as Protected DESIGNATION of origin. If so, please explain how „a particular type of dry-cured ham has begun to be produced in DIFFERENT Italian areas“ can be branded as San Daniele or Parma dry-cured hams. If not, please explain what Protected DENOMINATION of Origin stands for. To be clear, we just want to point out that seasoning for 13 months is just one of the requirements for Parma dry-cured hams. From what we can understand from this manuscript, if SSDCH hams produced in DIFFERENT Italian areas are to be seasoned for 13 months, they could be branded as San Daniele or Parma dry-cured hams.
Answer : Thanks I explain well in the abstract
No PDO San Daniele dry cured ham as well as PDO Parma hams must be produced in specific areas; the first in San Daniele and the second in Parma area. They can not be produced in other Italian area. To produce PDO dry cured ham you must apply the rules of the PDO: origin of meat, technology and times of seasoning. Both PDO Parma and San Daniele dry cured ham must be produced by pork leg, coming from pigs bred in the Italian peninsula, salt and spices and ripened over 13 months. Conversely not PDO dry cured ham can be produced using different pork leg coming from different Europe areas and ripened for less than 13 months.
So I added in the abstract that: Lines 19-25 - Recently, specific dry-cured hams have started to be produced in different Italian areas. The ingredients are similar to Protected Denomination of Origin (PDO) produced in San Daniele or Parma areas, and include pork leg, coming from pigs bred in the Italian peninsula, salt and spices. However these specific new products can not be marked as PDO either San Daniele or Parma dry cured ham, because they are seasoned for 6 months, and the mark PDO is given only to products seasoned over 13 months. Consequently these products are called short-seasoned dry-cured ham (SSDCH) and are not branded PDO.
#2: As authors have rightfully acknowledged, inadequate processing procedures can increase the development of moulds that can lead to negative effects, such as the production of mycotoxins (OTA-producing moulds or fungus). Then the authors also state that the use of short seasoning times can cause high moisture levels on the surface of the hams, which has detrimental effects, such as the growth of OTA-producing moulds. Our conclusion is that short seasoning times can be considered as “inadequate processing procedure”. Perhaps we should advocate that this economy (money) driven practise (shortening of time) leads to unsafe products, instead of finding the ways to use various agents (chemical, physical and biological) for food detoxification purposes or limitation and inhibition of OTA-producing moulds. More so because the authors also stated that HACCP system can reduce the level of OTA-producing mould contamination on SSDCH.
Answer: Thanks. It is possible that short ripening can bring to high OTA producing mould growth, but as demonstrated the use of a starter can limit their growth. I agree with you I think it is better to use long ripening to be sure to reach a real safety of the product. But the economy keeps not in mind that. However I just have previous experience that the checking of the technology also these products can be safe.
#3: Line 33: LAB species in red font ?!
Answer: Thanks I corrected it
Therefore, I would recommend the paper to be published after minor revisions.
